Measuring mass: variation among 3,161 species of Canadian Coleoptera and the prospects of a mass registry for all insects

Hu Jingchan 1 2
Pentinsaari Mikko 1 mpentins@uoguelph.ca
http://orcid.org/0000-0002-3081-6700 Hebert Paul D.N. 1 2
1 Centre for Biodiversity Genomics, University of Guelph , Guelph, Ontario , Canada
2 Department of Integrative Biology, University of Guelph , Guelph, Ontario , Canada
Gillespie Joseph
Electronic publication date: 2022 Jan 21
Publication date: 2022
Volume: 10
Electronic Location ID: e12799
Received 2021 Nov 3; Accepted 2021 Dec 24
Copyright: © 2022 Hu et al.
Copyright year: 2022
Copyright holder: Hu et al.
License: This is an open access article distributed under the terms of the Creative Commons Attribution License, which permits unrestricted use, distribution, reproduction and adaptation in any medium and for any purpose provided that it is properly attributed. For attribution, the original author(s), title, publication source (PeerJ) and either DOI or URL of the article must be cited.
License URL: https://creativecommons.org/licenses/by/4.0/

Keywords: Biomass, DNA barcoding, Coleoptera, Body mass

Funding: Ontario Graduate Scholarship Program to Jingchan Hu NSERC Discovery Grant to Paul D. N. Hebert 04797 and an infrastructure grant from the Gordon and Betty Moore Foundation to the Centre for Biodiversity Genomics (No. GBMF2966). The funders had no role in study design, data collection and analysis, decision to publish, or preparation of the manuscript This work was supported by a grant from the Ontario Graduate Scholarship Program to Jingchan Hu, an NSERC Discovery Grant to Paul D. N. Hebert (No. 04797), and an infrastructure grant from the Gordon and Betty Moore Foundation to the Centre for Biodiversity Genomics (No. GBMF2966). The funders had no role in study design, data collection and analysis, decision to publish, or preparation of the manuscript.

==============================
Although biomass values are critical for diverse ecological and evolutionary analyses, they are unavailable for most insect species. Museum specimens have the potential to address this gap, but the variation introduced by sampling and preservation methods is uncertain. This study quantifies species-level variation in the body mass of Canadian Coleoptera based on the analysis of 3,744 specimens representing 3,161 Barcode Index Number (BIN) clusters. Employing the BIN system as a proxy for species allows the inclusion of groups where the taxonomic impediment prevents the assignment of specimens to a Linnaean species. By validating the reproducibility of measurements and evaluating the error introduced by operational complexities such as curatorial practice and the loss of body parts, this study demonstrates that museum specimens can speed the assembly of a mass registry. The results further indicate that congeneric species of Coleoptera generally have limited variation in mass, so a genus-level identification allows prediction of the body mass of species that have not been weighed or measured. Building on the present results, the construction of a mass registry for all insects is feasible.

Introduction

Body mass is a key property of organisms which impacts factors ranging from metabolic rate to community structure, foraging behavior, and predator-prey dynamics (Peters, 1986; Chown & Gaston, 2010; Smith et al., 2016). Comprehensive body mass registries are available for mammals (Jones et al., 2009), fishes (Froese & Pauly, 2021), and birds (Dunning, 2008), but similar information is lacking for insects despite their abundance and ecological importance. Three characteristics of insects have impeded its assembly: (1) high species diversity, (2) variable curatorial practices, and (3) small size requires access to precision balances (Braun et al., 2009; Chown & Gaston, 2010; Gilbert, 2011; Knapp, 2012). These barriers to direct mass measurements have led many studies to employ estimates derived from body length, even for groups with very divergent body plans (Rogers, Hinds & Buschbom, 1976). Despite its lack of precision (Gowing & Recher, 1984; Johnston & Cunjak, 1999), this approach has been widely applied due to its simplicity (Ulrich, 2007; Greve et al., 2018; Richard, Tallamy & Mitchell, 2019). Aside from the fact that direct measurements of body mass for arthropods are uncommon, existing data are difficult to access because there is no structured data repository (Chown & Gaston, 2010).

The construction of a mass registry for insect species would benefit studies that currently depend on imprecise surrogates, facilitating comparisons across groups with differing morphology. Aggregating data from all insect orders and other arthropods, which are typically studied independently, would advance understanding of mass variation and its evolutionary trajectories across lineages (Ulrich, 2007; Chown & Gaston, 2010). Museum specimens has been proposed as a resource to assemble mass data for insects without new sampling effort (Gilbert, 2011). However, to assess the quality of the mass data resulting from their analysis, the impact of varying curatorial and preservation methods requires investigation.

With over 380,000 described species (Bouchard et al., 2017), Coleoptera is one of the most diverse orders of insects. Occurring in both aquatic and terrestrial environments, it includes some of the largest and smallest insects with its component taxa spanning eight orders of magnitude in mass (Chown & Gaston, 2010). These factors make Coleoptera an ideal group for developing approaches to support the construction of a mass registry for all insects. The present study targets the Coleoptera of Canada, a fauna of nearly 9,000 species (Brunke et al., 2019), many possessing a DNA barcode record on BOLD (Ratnasingham & Hebert, 2007). The mass data gathered in this study provide a strong basis for comparison with previous surveys (Chislenko, 1981; Novotny & Kindlmann, 1996; Ulrich, 2007). Also, because these values derive from specimens with DNA barcodes, it begins to develop the information on mass variation needed to advance both metabarcoding and eDNA analyses (Braukmann et al., 2019).

This study details variation in the body mass of 3,161 BIN clusters of Coleoptera based on the analysis of 3,744 museum specimens. It evaluates the impacts of humidity, tissue loss, and curatorial variables on mass. It also examines the extent of variation in mass among taxonomic lineages, work which indicates that phylogenetic constraints are strong enough for the current data to allow mass estimation for most Canadian Coleoptera. Finally, this study considers how best to expand from the current registry that includes records for a few thousand species to one with coverage for all insect species.

Materials and Methods

BINs as a species proxy

DNA barcoding employs sequence variation in a 658 base pair segment of the cytochrome c oxidase subunit I gene (COI) as a basis for specimen identification and species discovery in animals (Hebert et al., 2003). The BIN system clusters these COI sequences into molecular operational taxonomic units (Floyd et al., 2002) that correspond closely with Linnaean species. For example, about 90% of all species in the well-studied European Coleoptera fauna show perfect correspondence with BINs (Pentinsaari, Hebert & Mutanen, 2014; Hendrich et al., 2015). Divergent BINs associated with the same Linnaean species name often show morphological differences upon closer inspection, while some closely related species share BINs or even identical barcode haplotypes (Pentinsaari, Hebert & Mutanen, 2014; Hendrich et al., 2015). Each BIN is assigned a unique alphanumeric identifier that serves as a species proxy (Ratnasingham & Hebert, 2013). Because they provide a taxonomic assignment for undescribed species (Brunke et al., 2019; Pentinsaari et al., 2019; Brunke, Pentinsaari & Klimaszewski, 2021), this study employed BINs to structure data collection. While substantial efforts were also made to assign each BIN to a Linnaean species, this was not always possible because of both the lack of taxonomic specialists for some families and difficulties in resolving synonymies and cryptic species. As a result, we employ the BIN count as the best estimator of the number of species examined in this study.

Body mass data

Specimens were available for 3,161 BINs of Canadian Coleoptera. They represented 1,100 of the 2,008 genera (54.8%) and 96 of the 111 beetle families (86.5%) known from Canada (Bousquet et al., 2013). Most were morphologically identified to a genus (3,156 BINs) and many to a species (2,719 BINs to 2,389 recognized species). The specimens were obtained from sampling programs coordinated by the Centre for Biodiversity Genomics (CBG) at the University of Guelph and are stored in its voucher collection. Specimens missing major body segments (head, abdomen) were not weighed. In total, 3,744 specimens were analyzed, meaning a single specimen was weighed for most BINs. However, 2–8 specimens were weighed for 334 BINs, and for these taxa, a mean mass was calculated. Standard deviation and coefficient of variation were calculated for all BINs represented by at least three specimens in order to explore the extent of mass variation within BINs.

Specimens fell into three main curatorial categories (ethanol, pointed, pinned). Specimens from 70% ethanol were first air-dried and then weighed repeatedly until the mass measurement stabilized. Specimens on points were unmounted using 70% ethanol, dried, and weighed. Pinned specimens were weighed on their pin and the pin mass was subtracted. When mass variation among pins of the size used on a specimen exceeded 12.5% of its overall weight (Gilbert, 2011), it was unpinned and weighed directly.

The mass of small specimens (~2,800 representing 2,500 BINs) was quantified to the nearest 0.0001 mg using a high-precision balance (Mettler Toledo™ XP6U), while the ~950 larger specimens (>10 mg) were weighed to the nearest 0.1 mg using a less sensitive instrument (Mettler Toledo™ MS104S). Up to three significant figures were recorded. The BIN, taxonomic assignment, and mass of each analyzed specimen are provided in a supplementary document (Data S1).

In order to examine the relationship between fresh and dry body mass, 73 specimens of Coleoptera representing a range of body sizes were freshly collected, killed, and weighed within 24 h. The specimens were then stored dry at room temperature and weighed every 60 days until their mass stabilized, and the final weight was recorded as their dry mass. A linear regression was performed to assess the correlation between fresh and dry mass.

Data description and distribution analyses

All analyses were performed in R version 3.6.3 (R Core Team, 2020). Mass values were log-transformed before further analysis, and the interquartile range (IQR) was used as a measure of variance. A two-sided D’Agostino test was employed to evaluate skewness in the data, and an Anscombe-Glynn test to assess kurtosis. One-way ANOVAs using family, subfamily, and generic assignments as variables were used to assess mass variation at different levels in the taxonomic hierarchy using respective groups containing two or more quantified BINs. A nested ANOVA was also used to examine variation partitioning in the 50 families and 449 genera that were nested with two or more subgroups. Because taxonomic groups with low species diversity tend to show less variation in mass (Chown & Gaston, 2010), a separate analysis was performed to ascertain how variation in mass was partitioned in the most diverse groups of Canadian Coleoptera. In particular, a one-way ANOVA examined 65 genera with mass data for 10 or more BINs, while a nested ANOVA examined the six families with mass data for >100 BINs. To quantify the extent of mass that could be partitioned at each taxonomic level, ω2 values were calculated for all ANOVAs.

Pin variation and reproducibility assessment

A pinned insect is not easily separated from its pin, creating a complexity because the pin can outweigh the specimen. Gilbert (2011) proposed a workaround that involves estimating pin mass from key parameters (material, shape, size) before subtracting this value from the total weight to produce a mass estimate for the insect. Because the CBG employs insect pins from a single supplier, this source of variation was readily assessed. The mean and standard deviation in both diameter and mass was determined for 100 pins of each size. Because there was no overlap in diameter among different pin sizes, the size associated with each specimen could be determined, allowing its mass to be subtracted.

The consistency in determinations of pin size and of body mass was assessed by comparing mass values for 112 specimens weighed in 2014 and again in 2018. The congruence in net mass values was examined using a paired-sample t-test. To evaluate short-term variation in mass, 18 specimens were weighed daily on four consecutive days when variation in humidity was pronounced.

Since the specimens studied had been DNA barcoded before weighing, many lacked a leg or the part of it (e.g., tarsus) that was used for DNA extraction. As part of a set of pilot tests performed before commencing the weighing of specimens on a large scale, 50 specimens were examined to determine the reduction in mass caused by the loss of a leg.

Results

Mass variation in Canadian Coleoptera

Measurements of the 3,744 beetles representing 3,161 BINs revealed their mass varied by more than five orders of magnitude (0.0024–797 mg) (Fig. 1). Among BINs with a species assignment, Ptiliola kunzei (Ptiliidae, BOLD:ACI8875) and Ptiliolum fuscum (Ptiliidae, BOLD:AAM7677) possessed the lowest mass (0.0056 mg). However, six BINs in the same subfamily (Ptiliinae) weighed less, and a specimen identified to the genus Nanosella (BOLD:ADH5266) had the lowest mass (0.0024 mg). The largest species was Hydrophilus triangularis (Hydrophilidae, BOLD:AAQ2470) at 797 mg. The median mass of all species was ~1.3 mg, represented in the data by species such as Bembidion nitidum (Carabidae, BOLD:AAD2752) and Dichelotarsus piniphilus (Cantharidae, BOLD:AAH0933). Considering all BINs, the mass distribution approximated a lognormal distribution with strong kurtosis (z = −7.39, p = 1.49−13) but insignificant skew (z = 1.96, p = 0.051). The coefficient of variation for the 196 BINs with a minimum sample size of three specimens ranged from 0.02 to 0.975 (Data S1). The fresh and dry body mass of 73 beetle specimens freshly collected from the field showed a strong linear relationship (R2 = 0.97), with dry mass being approximately 49% of the fresh mass.

Figure 1 Distribution of the log-transformed body masses for 3,161 BINs of Canadian Coleoptera.

Much of the variation in mass among species was linked to their higher taxonomic assignments (family, subfamily, genus) (Table 1. a–c). In fact, ω2 values indicated that 90% of the mass variation could be explained by higher taxonomic placement with 55% of the variation at the family level, 20% at the subfamily level, and 15% at the genus level. Because of these relationships, variation in mass among congeneric species was typically limited (Fig. 2). Considering all 518 genera where two or more BINs were examined, the variance, measured by IQR, had a median of 0.163 log10 (mg), which is a 1.4-fold difference and translates to +/− 20% divergence from the median for the genus. Cases of extreme variation where the larger members of the genus were as twice as massive as the median (IQR > 0.6) were only observed in nine of the 518 genera, including Anotylus (Staphylinidae), Dyschirius (Carabidae) and Ilybius (Dytiscidae) (Data S1). Genera with ten or more measured BINs showing low IQR (<0.15) included e.g., Acrotrichis (Ptiliidae), Anaspis (Scraptiidae), and Paria (Chrysomelidae) (Data S1).

Table 1 Output of one-way and nested analyses of variance.

Analysis	Factor	D.f.	Sum Sq.	Mean Sq.	F	p	ω 2	
(a) One-way: family	Family	78	7,644	98.0	51.6	< 2.2−16	0.56	
Residual	3,068	5,821	1.9				
(b) One-way: subfamily	Subfamily	183	9,412	51.4	46.0	< 2.2−16	0.74	
Residual	2,749	3,075	1.12				
(c) One way: genus	Genus	518	9,934	19.2	50.0	< 2.2−16	0.91	
Residual	2,059	790	0.38				
(d) One way: genera with n ≥ 10	Genus	64	3,630	56.7	118.2	< 2.2−16	0.88	
Residual	957	459	0.48				
(e) Nested: family and genus levels	Family	49	6,185	126.2	329.6	< 2.2−16	0.59	
Family/Genus	448	3,561	7.95	20.8	< 2.2−16	0.32	
Residuals	2,013	771	0.38				
(f) Nested: families with n ≥ 100	Family	5	3,046	609.2	1407	< 2.2−16	0.51	
Family/Genus	269	2,383	8.9	20.5	< 2.2−16	0.38	
Residuals	1,174	508.4	0.4				

Figure 2 Distribution of the variance in body mass for Coleoptera genera.

Interquartile range measures the difference between the upper and lower quartiles and can be converted to fold-difference or used to estimate the typical deviation from the median (e.g., IQR 0.2 = 1.6-fold difference between quartiles ≈ +/− 23% from median; IQR 0.4 = 2.5-fold difference between quartiles ≈ +/− 57% from median).

Reliability of specimen mass measurements

Diameter measurements allowed the discrimination of each pin size as differences among pins of a particular size were an order of magnitude (±0.005 mm) less than the diameter difference (0.05 mm) between adjacent pin sizes. Pins of one size did vary in mass (±0.2–0.9 mg) with this variation increasing with larger pins, but it usually represented a small component of the overall mass. In the few cases where variation in pin mass represented >12.5% of the total weight, the specimen was unmounted and weighed directly.

High humidity slowed analysis as the balances required longer to stabilize, but changes in mass linked to variation in temperature and humidity were small. For example, the mean standard error based on measurements of 18 specimens weighed every 24 h over four days was 1–2% of their mean (Data S1). Comparison of specimen weights between 2014 and 2018 further indicated that differences between paired measurements were within ±3% of their average in 109 of 112 cases while the others were within ±5%. A paired t-test demonstrated that mean mass increased by 0.3% (t = 2.0657, df = 111, p = 0.02) over the interval, likely reflecting higher humidity when the second measurements were made.

Based on a pilot test conducted before weighing specimens for analyses (full data not available), the loss of an appendage had a small impact on mass. For example, a leg typically represented 1–2% of the specimen’s mass, while the tibia plus tarsus were around 0.5%. The loss of a major body segment (head, abdomen) had much larger impacts as they comprised 12–50% of the total mass.

Discussion

Because of its strong association with crucial biological traits, mass data is valuable in many ecological and evolutionary contexts (Niven & Scharlemann, 2005; Beukeboom, 2018). By assembling mass data for 3,161 Coleoptera BINs, this study confirms that museum specimens are a valuable resource for constructing a mass registry. It further demonstrates that factors such as the loss of an appendage, variation in humidity, specimen age, and curatorial practices have small impacts on these measurements.

This study further demonstrates that mass variation among beetles has strong phylogenetic constraints with much of the variation residing at the family, subfamily, and generic levels. As a consequence, the analysis of a single or a few individuals of a species provides a good estimate of its mass. Prior studies have demonstrated that adult body size can be impacted by diverse environmental factors and by sexual dimorphism, and that the extent of such variation differs among species (Emlen & Allen, 2003; Kawano, 2006; Chown & Gaston, 2010; Tseng et al., 2018). For example, many species of Scarabaeidae and Lucanidae show both prominent sexual dimorphism and extensive size variation within sexes (Kawano, 2006). As extreme variability of adult body size due to variation in the quantity and quality of larval food has also been well documented in e.g., Cerambycidae (Svacha & Lawrence, 2014), it was not surprising that 15 of the 30 BINs with the highest coefficients of variation belonged to this family. These impacts can even shift the relationship between morphometric measurements (e.g., body length) and biomass (Gouws, Gaston & Chown, 2011). The inclusion of multiple representatives of both sexes in mass databases is warranted in such taxa to provide a more accurate estimate of mean body mass, to account for this variation in analyses, and to improve the accuracy of body size prediction of congenerics. However, given the millions of insect species, investigating such impacts on a species-by-species basis is not feasible, and it is unnecessary in most contexts because they generally cause only minor modulations in body mass as >90% of variation resides at higher taxonomic levels (Chown & Gaston, 2010). While a single data point per species does not provide a detailed characterization of its mass, the collective data for a genus or subfamily are a meaningful indicator of body size in the lineage.

Because the present analysis only considered adult body mass, these data will be of lesser value for samples consisting largely of larvae as their mass increases by orders of magnitude during development (e.g., Nijhout, Davidowitz & Roff, 2006). In fact, mature larvae of holometabolous insects frequently outweigh adults (Waters & Harrison, 2012). Adult body mass is more useful in many modeling situations (e.g., female fecundity typically shows a positive correlation with body mass (Taylor, Anderson & Peckarsky, 1998; Beukeboom, 2018)).

Importantly, mass estimates derived from generic assignments are similar in accuracy to those resulting from the standard approach to mass estimation: the use of a power equation to estimate mass from body length (Rogers, Hinds & Buschbom, 1976). While our data set is larger and spans a greater range of mass, our residual SE (0.62) was less than that resulting from the use of a power equation (Table 1.c). In fact, even when our analysis targeted genera with the most variation in mass among their component species, the residual SE (0.69) was similar to that (0.66) reported with the use of a power equation (Rogers, Hinds & Buschbom, 1976) (Table 1.d). While this residual translates into an average two-fold difference in mass from the predicted value, it indicates that a generic assignment can generate mass estimates with a precision similar to those based on estimates from direct length measurements. In short, mass values for a few species in each genus allow the estimation of mass for congeneric taxa. Although our results only document this fact for Coleoptera, similar relationships undoubtedly extend to other groups, as strong phylogenetic signal in body size occurs in many arthropod lineages (Rainford, Hofreiter & Mayhew, 2016). Understanding the extent of phylogenetic constraint in arthropods could greatly speed the development of a functional mass registry by allowing analysis to focus on groups where size variation is most pronounced and to use proxy measures in those where it is not.

By delivering information on body mass for about 0.2% of all described insect species, the present study indicates that it is feasible to construct a mass registry for all insects. Furthermore, it reveals shortcuts to develop this registry. Specifically, the strong phylogenetic constraints on mass indicate that early efforts should focus on gaining coverage for higher taxonomic categories–every insect family and subfamily. Work should then extend to every genus and in time to every species. Because this effort will generate a substantial volume of data, it needs a home and the BOLD platform (Ratnasingham & Hebert, 2007) is well-suited to meet this need. Although specimens with mass data need not possess barcode records, the inclusion of sequence information will maximize the utility of these records for metabarcoding and eDNA analysis: As larger specimens tend to release more DNA, information on body mass is crucial for estimating abundances from bulk samples, although the impacts of DNA extraction and PCR amplification protocols must also be considered (Elbrecht, Peinert & Leese, 2017; Deagle et al., 2018).

Moreover, the barcode records will ensure that specimens in the mass registry are properly identified, one of the key problems confronting any large-scale repository of biological collaterals. To demonstrate its capacity, the current records are deposited in a dataset on BOLD (DS-MASSCOL; dx.doi.org/10.5883/DS-MASSCOL) that couples barcode records with mass information on each specimen examined in this study. Because DS-MASSCOL is a dynamic dataset where BIN assignments may shift and where specimens that currently lack a genus or species assignment may gain one, a Supplemental File (Data S2) has been provided as a snapshot at the time of submission.

Conclusions

Aside from its value on providing a basis for extending understanding of the evolution of body mass, comprehensive body mass data on insect species is needed for ecological modeling. By confirming that variable curatorial and preservation practices have little impact on body mass, the present study establishes that museum specimens provide a resource for the rapid assembly of mass data. Employing this approach, the present study assembled mass data for 3,161 species of Coleoptera, nearly 1% of all known species in this order. Moreover, because of the strong phylogenetic constraints on body size, the current records enable accurate mass estimation (+/−20%) for nearly all Canadian beetles. The extension of this approach to other arthropod groups and other geographic regions would facilitate the assembly of a mass registry for all insects. Incorporation of the resultant mass value for each BIN into the parameters on BOLD will ensure easy access to these data.

Supplemental Information

Supplemental Information 1 Raw mass data for each specimen and summaries of mass variation within BINs and genera.

Raw mass measurements for each specimen in the dataset; Mass data summarized by BIN cluster; Mass variation within BINs; Interquartile range by genus; Mass measurements repeated in 2014 and 2018; Daily variation in specimen mass over 4 days; Fresh vs dry mass in 73 specimens of Coleoptera.

Click here for additional data file.

Supplemental Information 2 Snapshot of the BOLD dataset DS-MASSCOL at the time of submission.

All available collection data, full taxonomic data, and other information related to the analyzed specimens as available on BOLD at the time of submission. Body mass measurements are stored in a custom field generated for this purpose (tab: Custom).

Click here for additional data file.

Supplemental Information 3 R code file containing the scripts used in the statistical analyses.

Click here for additional data file.

Additional Information and Declarations

Competing Interests

Author Contributions

Data Availability

The authors declare that they have no competing interests.

Jingchan Hu conceived and designed the experiments, performed the experiments, analyzed the data, prepared figures and/or tables, authored or reviewed drafts of the paper, and approved the final draft.

Mikko Pentinsaari conceived and designed the experiments, analyzed the data, authored or reviewed drafts of the paper, and approved the final draft.

Paul DN. Hebert conceived and designed the experiments, analyzed the data, authored or reviewed drafts of the paper, and approved the final draft.

The following information was supplied regarding data availability:

The body mass measurements for each individual specimen, the mean mass for BINs (proxy for species) with multiple specimens analyzed, a summary of mass variation within BINs, and raw data for a set of smaller scale experiments reported in this study; a snapshot of the BOLD dataset at the time of submission, including full taxonomic data as well as all available data on specimen collection and storage; and the R code for the analyses are available in the Supplemental Files.

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
