# Peer review of "Measuring mass: variation among 3,161 species of Canadian Coleoptera and the prospects of a mass registry for all insects"

_PeerJ, doi:10.7717/peerj.12799_

## Round 0.1 · original submission · Major Revisions

Dear Dr. Hu and colleagues:

Thanks for submitting your manuscript to PeerJ. I have now received two independent reviews of your work, and as you will see, the reviewers raised some concerns about the research. Despite this, these reviewers are optimistic about your work and the potential impact it will have on research studying Canadian Coleoptera. Thus, I encourage you to revise your manuscript, accordingly, taking into account all of the concerns raised by both reviewers.

Reviewer 1 kindly provided feedback on a marked-up version of the manuscript.

Please provide all of the missing references pointed out by the reviewers.

There seems to be some comparisons that can be carried out that will make your work more relevant to previous morphological determinations.

Please ensure that all scripts and related tools are available to the public.

There are many comments by both reviewers that ask for more information on specific issues; please address these.

While the concerns of the reviewers are relatively minor, this is a major revision to ensure that the original reviewers have a chance to evaluate your responses to their concerns.

I look forward to seeing your revision, and thanks again for submitting your work to PeerJ.

Good luck with your revision,

-joe

·

Basic reporting

In this study Hu et al. quantify the variation of body mass for more than 3,100 beetle species from Canada, employing the BIN approach based on DNA barcodes for their discrimination. In my eyes, the topic of this manuscript is interesting and appropriate for “PeerJ”, but I feel that some aspects should be pointed out/discussed more in detail.

The authors claim that “… factors such as the loss of an appendage, variation in humidity, specimen age, and curatorial practices have small impacts on these (mass) measurements.” I think it would be nice to provide some examples of a direct comparison of old and fresh specimens, at least for some selected abundant species. I do not see the necessity to rely for such studies on the existence of DNA barcodes for the analyzed specimens if the identification/classification is without doubt.

Furthermore, strong mass and size variations are well-known for numerous families and genera, e.g., the genera Pterostichus (Carabidae) or Otiorhynchus (Curculionidae), whereas other genera have almost no size/mass variation (e.g., Nothiophilus or Dyschirius (both Carabidae)). I feel that the authors should present some specific examples of their results.

Phenotypic plasticity is a well-known phenomenon for a variety of beetle species, e.g., for stag beetles (Lucanidae), and can affect mass and size of specimens significantly. A discussion of this aspect is still missing but should be added.

Finally, I feel that there is a good chance to increase the number of relevant references.

Please check the uploaded pdf for details also.

Experimental design

The research question is well explained, the used methods adequate. Many data, however, rely only on one analyzed specimen per species. What can you say about the intraspecific variation for the studied species with more than one specimen/species? I think these data are important to show.

Validity of the findings

As already pointed out I think that the topic of this manuscript is interesting and appropriate for “PeerJ”. However, I feel that the authors should present some specific examples of their results.

Additional comments

Maybe the authors can provide some images of pinned specimens and/or insect boxes for illustration.

Reviewer 2 ·

Basic reporting

DNA barcoding reference libraries are critical to help species identification, especially for hyperdiverse taxa. However, vouchers associated metadata are often overlooked while they may be used and of importance in ecological studies. Linking biological traits like body mass to these reference libraries can be helpful to further use DNA (meta)barcoding in ecological and functional studies rather than biodiversity inventories only. In that sense, this paper is a real effort to improve associated DNA barcoding reference libraries metadata and a great contribution to the field overall. While the main pitfall is the huge number of singletons used to mass species proxy, the work remains of a great scale and has to be acknowledged. The work around curatorial categories is also of interest to standardize mass measurements from collection specimen.

In that sense, I believe that the article fits the scope of PeerJ, with a research being sound and of potential interest to a broad audience.

Main comment 1:
I have been a bit surprised to learn only L. 96 (in the Material and Methods) that the 3,161 species are actually 3,161 BINs. I fully agree and acknowledge the use of BINs as species proxy, but this should be clarified earlier (abstract and introduction) in the manuscript. Mentioning directly BINs as species proxy or genetic species under consideration would clarify the manuscript and remove doubt to the reader, as BINs are not always following Linnaean delineation. Indeed, you show L. 103 that your actual BINs with species level (2,719) correspond to 2,389 recognized species. This is 330 SPLIT cases that actually inflate your species count when considering BINs as species. Please clarify your manuscript accordingly, when using BINs as species proxy or recognized species (See secondary comments 1 and 3, but also title, abstract and all manuscript parts).

Main comment 2:
One issue I face in the manuscript is the huge number of singletons considered for weighting BINs (3,744 individuals for 3,161). I acknowledge the huge “sampling” effort it took to weight all these individuals to generate mass data. Actually, I believe that the number of individuals should be earlier mentioned in the manuscript (L. 23 in Abstract and 73. in Introduction) to better emphasized the impressive work here. However, I do lack more discussion on the intraspecific variability that can arise within species. Paragraph L. 220 – 229 nicely discuss interspecific mass variations, but nothing is said about the intraspecific variation. Considering that sex, resources availability, reproduction strategies, etc are important factors that can make interindividual mass vary greatly (potentially even greater than some interspecific variations), please acknowledge and discuss this shortfall and that further work should be done to complement the database in that sense.

Main comment 3:
You mention L. 111 that you had three main curatorial categories (ethanol, pointed, pinned). I was wondering if you had BINs with multiple curatorial categories available (through multiple individuals) to also compare between them directly?

Main comment 4:
Sentence L. 69–71 is important. I believe that the article would gain in importance if shortly discussing the implication of this work for such techniques (e.g. to help inferring relative abundance for example).

Experimental design

no comment

Validity of the findings

no comment

Additional comments

Secondary comment 1:
L. 87–88: Please nuance a bit the sentence as the BIN system clustering does not always correspond at 100% with Linnaean species.

Secondary comment 2:
L. 201: Please provide full T-test parameters (t = XX; df = XX; p-value = XX).

Secondary comment 3:
L. 212: If not clarified earlier in the manuscript that BINs are used as species proxy, please clarify “3,161 species of Coleoptera”. (See main comment 1.)

Comment on supplementary data availability and reproducibility:
I thank you for publishing the dataset with a proper DOI. However, I lack R scripts for the analyses performed. For the sake of reproducibility, could it be possible to make it available on GitHub or other public and dedicated repository, along with the data used (e.g. humidity, weights in 2014 and 2018, weight of appendage, etc), with a link or DOI mentioned in the appropriate section of the manuscript?

---

## Round 0.2 · accepted · Accept

Dear Dr. Hu and colleagues:

Thanks for revising your manuscript based on the concerns raised by the reviewers. I now believe that your manuscript is suitable for publication. Congratulations! I look forward to seeing this work in print, and I anticipate it being an important resource for groups studying Canadian Coleoptera. Thanks again for choosing PeerJ to publish such important work.

Best,

-joe